# Multidisciplinary Collaboration for the Optimization of Antibiotic Prescription: Analysis of Clinical Cases of Pneumonia between Emergency, Internal Medicine, and Pharmacy Services

**DOI:** 10.3390/antibiotics11101336

**Published:** 2022-09-30

**Authors:** Lorea Arteche-Eguizabal, Iñigo Corcuera-Martínez de Tobillas, Federico Melgosa-Latorre, Saioa Domingo-Echaburu, Ainhoa Urrutia-Losada, Amaia Eguiluz-Pinedo, Natalia Vanina Rodriguez-Piacenza, Oliver Ibarrondo-Olaguenaga

**Affiliations:** 1Osakidetza Basque Health Service, Debagoiena Integrated Health Organization, Pharmacy Service, 20500 Arrasate/Mondragón, Spain; 2Osakidetza Basque Health Service, Debagoiena Integrated Health Organization, Internal Medicine Service, 20500 Arrasate/Mondragón, Spain; 3Osakidetza Basque Health Service, Debagoiena Integrated Health Organization, Emergency Service, 20500 Arrasate/Mondragón, Spain; 4Osakidetza Basque Health Service, Debagoiena Integrated Health Organization, Research Unit, 20500 Arrasate/Mondragón, Spain; 5Biodonostia Health Research Institute, 20014 Donostia-San Sebastián, Spain

**Keywords:** pneumonia, antibiotic stewardship, emergency, internal medicine, pharmacy

## Abstract

Background: Pneumonia is a lung parenchyma acute infection usually treated with antibiotics. Increasing bacterial resistances force the review and control of antibiotic use criteria in different health departments. Objective: Evaluate the adequacy of antibiotic treatment in community-acquired pneumonia in patients initially attended at the emergency department and then admitted to the internal medicine service of the Alto Deba Hospital—Osakidetza Basque Country Health Service (Spain). Methods: Observational, retrospective study, based on the review of medical records of patients with community-acquired pneumonia attended at the hospital between January and May 2021. The review was made considering the following items: antimicrobial treatment indication, choice of antibiotic, time of administration of the first dose, adequacy of the de-escalation-sequential therapy, duration of treatment, monitoring of efficacy and adverse effects, and registry in the medical records. The review was made by the research team (professionals from the emergency department, internal medicine, and pharmacy services). Results: Fifty-five medical records were reviewed. The adequacy of the treatments showed that antibiotic indication, time of administration of the first dose, and monitoring of efficacy and adverse effects were the items with the greatest agreement between the three departments. This was not the case with the choice of antibiotic, de-escalation/sequential therapy, duration of treatment, and registration in the medical record, which have been widely discussed. The choice of antibiotic was optimal in 63.64% and might have been better in 25.45%. De-escalation/oral sequencing might have been better in 50.91%. The treatment duration was optimal in 45.45% of the patients and excessive in 45.45%. Discussion: The team agreed to disseminate these data among the hospital professionals and to propose audits and feedback through an antibiotic stewardship program. Besides this, implementing the local guideline and defining stability criteria to apply sequential therapy/de-escalation was considered essential.

## 1. Introduction

Community-acquired pneumonia (CAP) is an acute disease caused by an infection of the lung parenchyma acquired outside of a hospital setting. It is one of the leading global causes of morbidity and mortality in immunocompetent and immunocompromised [1], with direct relation to patient’s comorbidities. Consequently, CAP has a relevant impact on vulnerable populations’ health, causing high costs for the health systems. CAP is also conditioned by other determinants such as socioeconomic status with a bigger effect on low socioeconomic populations [2].

The incidence is very variable across different countries. In Europe, CAP incidence varies from 79.9/10,000 person-years in the UK [3], 47/10,000 person-years in France [4] to 20.6/10,000 in Iceland [5]. In Spain, pneumonia is not a reportable disease and therefore the incidence is unknown. However, the estimated incidence in adults in the Spanish primary care setting between 2009 and 2013 was 4.63 cases per 1000 persons/year, and it was higher with age and also, in males [6]. From 2016–2019, the average incidence of CAP was 1.2–3.5 cases per 1000 persons per year [7]. The risk factors related to CAP were, among others, HIV infection, chronic obstructive pulmonary disease, asthma, and smoking [6]. In the Basque Country (northern area of Spain) the incidence was higher in urban areas (6.62 cases per 1000 persons/year) [7].

The treatment of CAP is based on antibiotic administration, an effective strategy that reduces mortality and complications. The use of antibiotics has grown fast and the non-adequacy of targeted therapies against disease-causing microorganisms on many occasions, with a big ecological impact. At the same time, the resistance of pneumonia-causing bacteria to antibiotics has grown fast over the world. The Center for Disease Control and Prevention (CDC) establishes different infectious diseases in which there is an “overprescription” of antibiotics, including respiratory tract infections. This situation has forced a change of positions regarding the antibiotic treatment of CAP [8]. Therefore, the clinical management of CAP has become a real challenge for physicians [9].

In emergency departments, for example, there is the need for active use of specific tools as guidelines for the prescription election [10] and more efforts need to be done to promote and improve appropriate antibiotic usage, optimizing patient care [11]. Thus, the combination of audit and feedback plus distribution of a pocket version of guideline recommendations as antibiotic selection tools led to a substantial increase in appropriate empirical antibiotics prescription in a study carried out in a Norwegian hospital [12].

Numerous studies demonstrate that multidisciplinary collaborative interventions can improve antimicrobial prescription and clinical outcomes too: selecting the appropriate empiric antimicrobial therapy, providing rapid de-escalation of antimicrobial agents, and ensuring the correct duration of treatment [13,14]. There are different experiences with positive results at various levels: decrease in the prescription of antibiotics, decrease in prescriptions considered inappropriate, decrease in length of stay (LOS), reduction in mortality, increase in the adequacy of treatment, decrease in exposure time to broad-spectrum antibiotics, decrease in the number of patients treated with 1 or more excessive doses of antibiotics, decrease in the use of fluoroquinolones, increase in the percentage of patients treated for healthcare-associated pneumonia with antibiotics that are recommended in guides, among others. The alone training is not likely to lead to sustained change in optimizing antibiotic prescription when compared to training and multidisciplinary collaboration [14].

## 2. Objective

The objective of the present research work was to evaluate the adequacy of antibiotic treatment in community-acquired pneumonia in patients initially attended in the emergency department and then admitted to the internal medicine service of the Alto Deba Hospital—Osakidetza Basque Country Health Service (Spain). The analysis constitutes the first step to promoting improvements in the optimization of antibiotic prescription in hospital services.

## 3. Materials and Methods

The analysis was performed at the Debagoiena Integrated Health Organization (IHO), integrated at Osakidetza Basque Health Service in the Basque country in the north of Spain. The IHO is composed of one acute care hospital with 72 beds capacity and 4 Primary Care Units (PCU). The IHO has an assistance influence area of 69,000 inhabitants. We conducted a retrospective observational study considering patients diagnosed with pneumonia from January to May 2021 because they are the months with the highest incidence of pneumonia in the Basque country. The collected data were age, sex, diagnostic ICD code (International Classification of Diseases) with name, and antibiotic treatment. The treatment registration included antibiotic active ingredient, dose, frequency, total daily dose, total days of treatment, and urinary antigen test (antigenuria). It was unable to obtain sputum culture results.

On one hand, the CAP identification was made considering diagnostic registered by the services, using ICD9 and ICD10 codification. The ICD9 pneumonia code is 486, and the ICD10 codes are J18.9, J69.0, J15.9, J12.81, and J13. On the other hand, the pharmacological treatment identification was made using the codes of the anatomical-therapeutic-clinical classification (ATC): J01 subgroup to identify antimicrobials for systemic use.

The severity of the pneumonia was evaluated by Pneumonia Severity Index (PSI), one of the most frequently used scores in pneumonia severity classification. PSI is composed of 20 items and classifies patients into five categories of severity that are associated with the risk of mortality. Age and comorbidities are highly weighted in the PSI, and for these reasons, PSI can underestimate the severity of pneumonia in young patients and those without previous diseases [15]. The comorbidities of the patients were evaluated using the usual Charlson Index [16,17].

On the other hand, the evolution of the consumption of antibiotics was evaluated considering the defined daily dose (DDD), defined in the Appendix A).

### 3.1. Treatment Revision Procedure

The process of evaluating the medical records of identified patients in the period was carried out retrospectively. The review was carried out by health professionals comparing treatments with drug treatment guidelines. The evaluators followed an agreed process to determine the adequacy of the treatment and its characteristics. Thus, the evaluation team included professionals from different departments (emergency, internal medicine, and pharmacy). Employment of interprofessional and simulation pedagogies allowed the exploration of the contextual factors that affect the safe and effective management of infections [18]. The evaluation criteria were based on the study of prevalence and adequacy of hospital use of antimicrobials in Spain—PAUSATE [19] (in which study several researchers from the team participated). The considered treatment criteria according to disease severity, individual antibiotic, and treatment time period are shown in Table 1. Concordance of prescribing with the guidelines was defined as the correct selection of both the antibiotic agent and the route of administration. Thus, for example, if the antibiotic was administered intravenously for a PSI class I or II patient, prescribing was considered discordant [20].

### 3.2. The Items and the Evaluation Criteria Were:

#### 3.2.1. Indication of Antimicrobial Treatment

**Adequate:** There was a confirmed infection or a reasonable probability that the patient had one.

**Inadequate:** There was no infection or a reasonable probability that the patient had one.

**Doubtful:** Not enough elements of judgment are available.

#### 3.2.2. Choice of Antimicrobial Agent

**Optimal:** The antimicrobial agent was the one of choice in the local guide or, failing that, in the national reference guide or, failing that, at the discretion of the evaluator.

**Might be better:** There was a more suitable alternative, but the prescribed agent was effective in curing the infection.

**Inadequate:** The antimicrobial agent was insufficient to cure or prevent the infection.

**Doubtful:** Not enough elements of judgment are available.

Note: If a patient received more than one antimicrobial, consider the global treatment, not each agent separately (extensible to the other dimensions or items)

#### 3.2.3. Timing of Administration of the First Dose

**Adequate:** In sepsis/septic shock within 1 h from the onset of symptoms. In severe infection within 6 h of onset of symptoms or arrival at the hospital.

**Inadequate:** the times described above were not met

**Doubtful or unknown:** The time of administration of the antibiotic is not recorded

#### 3.2.4. Dosage and Frequency of Administration

**Optimal:** The dose and frequency of administration were those reflected in the local or reference guidelines and/or were adapted to the severity of the infection and the patient’s conditions.

**Might be better:** The dose and frequency of administration were effective in curing the infection but may cause minor problems related to its use (e.g., no dose adjustment in renal failure with less risk of toxicity).

**Inadequate:** The dose and route of administration were ineffective in curing the infection and/or may cause major problems related to its use (e.g., insufficient dose with risk of inefficacy, or excessive dose with greater risk of toxicity…).

**Doubtful:** Not enough elements of judgment are available.

#### 3.2.5. De-Escalation/Oral Sequencing

**Optimal**: The route of administration was that reflected in the local or reference guidelines and/or was adapted to the severity of the infection and the patient’s stability conditions.

**Might be better:** The route of administration was effective in curing the infection but could have caused minor problems related to its use (e.g., no oral route if indicated…).

**Inadequate:** The route of administration was ineffective in curing the infection and/or may cause major problems related to its use (e.g., oral route in severe infection)

**Doubtful:** Not enough elements of judgment are available.

#### 3.2.6. Treatment Duration

**Optimal:** Duration recommended in the local guide, or, failing that, in the national reference guide or, failing that, at the discretion of the evaluator.

**Excessive:** Duration at least 2 or more days than recommended.

**Short:** Duration at least 2 days less than recommended.

**Doubtful or unknown:** Not enough elements of judgment are available.

#### 3.2.7. Monitoring of Efficacy and Adverse Effects

**Adequate:** All pertinent actions have been carried out to control the efficacy and safety of the antimicrobial treatment (e.g., control of fever, leukocytosis, biomarkers if applicable; withdrawal of control blood cultures in bacteremia due to S. aureus; complete blood count in treatment with linezolid; creatinine serum in treatment with aminoglycosides and vancomycin…).

**Inadequate:** Not all pertinent actions have been carried out to control the efficacy and safety of the antimicrobial treatment Inadequate.

**Doubtful:** I have doubts or do not have sufficient elements of judgment.

Registration in the medical record

**Complete:** The justification, changes, and suspension of antimicrobial treatment are well documented in the medical record.

**Partial:** The justification, changes, and suspension of antimicrobial treatment are partially documented in the medical record.

**Lacking:** In the medical record there is no mention of antimicrobial treatment.

### 3.3. Statistical Analysis

A descriptive analysis of the variables was carried out to summarize the results of the evaluation of the treatment records. Association or differences between variables were assessed using the chi-square (Χ^2^) test or Fisher exact test as appropriate. All statistical analyses were done using R-Statistic (v. 4.1.3). The significance level was set at 95%.

## 4. Results

The analysis of antibiotic use in the years prior to the period studied showed that the prescription of antibiotics decreased in 2020 in the emergency and internal medicine departments by 16.46% and 4.8%, respectively, compared to the previous year. Even so, the relative consumption by DDD/100 admission, DDD/100 beds in internal medicine, and DDD/1000 visits in emergency service was greater than the previous year, Appendix A.

In the months of January to May 2021, 92 patients diagnosed with pneumonia were treated, 57 of which remained alive in May 2022, when the study was carried out. Two patients did not actually have pneumonia and were removed from the study. Finally, 55 patients were considered in the study. The mortality rate is 38.89% with a mean age of 87.74 years (CI: 85.52–89.96 years) in dead patients. The reviewed patients from emergency and internal medicine showed bigger gender distribution for males (60%) than females (40%). The mean age was 65.07 years (59.09–71.05). In men, the mean age is 66.06 and in women is 69.58. The microorganism distribution showed that *Streptococcus pneumoniae* antigenuria has been positive in 18.18% of patients, *Legionella antigenuria* has been negative in 92.73%, and in the rest of the admissions, specific identification had not been requested. The registered pneumonia severity and comorbidity index (Table 2) could not be identified in 16.36% of cases (it was not identified). The identified patients with the Charlson comorbidity index were between 0 to 3 in 72.72% of the patients. Otherwise, half of the patients had a registered PSI scale index between levels I to III. The administration procedure was intravenous in 100% of cases for patients with PSI I-II and 73.68% (14 patients) were discordant with the recommended agent (in PSI I-II too). The distribution of antibiotics administered to treated patients is shown in Figure 1. Ceftriaxone, Levofloxacin, and Azithromycin were the antibiotics with the highest number of prescriptions.

The results on the adequacy of the treatments (Table 3) showed that antibiotic indication, time of administration of the first dose, and monitoring of efficacy and adverse effects were the items with the greatest agreement between the three departments. This was not the case with the choice of antibiotic, de-escalation/sequential therapy, duration of treatment, and registration in the medical record, which have served to be widely debated. The choice of antibiotic was optimal in 63.64% and might have been better in 25.45%. De-escalation/oral sequencing might have been better in 50.91%. The registration in the medical record was complete only in 34.55% of the patients. Finally, the treatment durations according to the PSI scale are shown in Table 4. The treatment variability is much greater in patients with lower PSI (I or II). Patients with PSI-III show the shortest durations of treatment, although they correspond to the smallest sample of the available patient sample. The distribution of treatment time adequacy is shown in Appendix A. The treatment duration was optimal in 45.45% of the patients and excessive in 45.45%. The reasons for this excess are the fear that it will not be enough or because of the seriousness of the condition.

## 5. Discussion/Conclusions

The selection of the antibiotic treatment is conditioned by multiple factors, becoming the management cornerstone of patients with pneumonia. There are a limited number of randomized clinical studies addressing antimicrobial stewardship strategies in CAP, but some possible changes have been proposed. Some studies have indicated that empirical treatment with beta-lactam monotherapy is non-inferior to a beta-lactam-macrolide combination or fluoroquinolone monotherapy with regard to 90-day mortality [21]. In the same way, oral drug switching is safe and effective. Thus, a 3-step critical pathway treatment ((1) early mobilization of patients; (2) use of objective criteria for switching to oral antibiotic therapy; and (3) use of predefined criteria for deciding on hospital discharge) reduced the duration of intravenous antibiotic therapy and did not adversely affect patient outcomes [22]. Thereby, antibiotic treatment withdrawal based on clinical stability criteria after a minimum of 5 days of appropriate treatment is not inferior to traditional treatment schedules in terms of clinical success [23]. For patients admitted to the hospital with CAP, who met clinical stability criteria, discontinuing β-lactam treatment after 3 days was non-inferior to 8 days of treatment [24]. Three main strategies in optimizing antibiotic treatment have been proposed [25]: (1) improve the documentation on the indication of antibiotics, the days of treatment, and the expected duration; (2) improve the accessibility of the current guidelines for the most common infections; and (3) perform a 72 h antibiotic “time-out” after the onset of symptoms to determine the adequacy of antibiotic coverage.

Other criteria, such as de-escalation, identifying inappropriate antibiotics with the microorganism, unlikely infection, inadequate double coverage, and sequential therapy, can be singled to optimize antibiotic prescription [26]. For all this, there is evidence that antibiotic stewardship initiatives can be securely applied: antibiotic de-escalation, duration of antibiotic treatment, adherence to CAP guidelines recommendations about empirical treatment, and switching from intravenous to oral antibiotic therapy may each be relevant in this context. The use of procalcitonin may be useful to improve antibiotic use but more evidence is needed [27,28]. The antibiotic consumption control needs the application of a specific antibiotic stewardship program (ASP). These action plans consider the control of the defined daily dose (DDD). It is defined as the average daily maintenance dose of a drug used for its main indication in adults. The number of DDD (DDD) is, at present, the most widely used unit since it allows comparisons to be made in the hospital globally or by services in a more generalized way, although it is not exempt from important limitations. DDD/100 beds and 100 admissions are classically used, which are intended to indicate the potentially exposed population, and both can be complementarily used [28,29]. The antibiotic use requires to be adjusted per 100 bed-days, and correlated with the case mix index, to better reflect the ASP activities, and to allow benchmarking with other organizations [30]. Taking these criteria into account, it has been observed that the consumption of antibiotics by several hospital-admitted patients has increased in our organization (Appendix A).

The trend in the use of antibiotics has changed in recent years, showing an increase in the defined daily dose (DDD/100 beds, DDD/100 admissions, and DDD/1000 visits) in the year prior to the considered time period in the study. Therefore, it is necessary to control the use of antibiotics and their suitability to reverse the trend. It is not only necessary to reduce the number of inadequate prescriptions but also treatment management. Follow-up after the initiation of the antibiotic treatment is also important, and the correct management should include early shifts to oral antibiotics, stewardship according to the microbiological results, and short-duration antibiotic treatment that accounts for the clinical stability criteria [15]. Thus, the indication of the antibiotic treatment was adequate in most cases, inappropriateness being minimal. This finding contrasts with other studies that found that 50% (30–60%) of the prescription of antibiotics were adequate [31,32,33]. In addition, antibiotic choice was inappropriate or doubtful in 10.91% of cases. These results are similar to other studies collected in the literature, which indicate that the choice of antibiotic was changed in 48 h in 16.77% of cases (107 patients in a group of 638) [34].

In our study, the de-escalation or oral sequencing was optimal in less than half of the cases, although antibiotic de-escalation seems to be safe and effective in reducing the duration of LOS and did not adversely affect outcomes of patients with community-acquired pneumococcal pneumonia, even those with bacteremia and severe disease, and those who were clinically unstable [35]. This area should be reviewed to increase the de-escalation procedures since de-escalation has no inferior results to the continuation of empirical treatment in patients with pneumonia [36]. Carugati et al. [37] indicated that de-escalation was not associated with an increased risk of either 30-day mortality or clinical failure during hospitalization. Furthermore, de-escalation did not seem to lead to higher 30-day mortality and clinical failure rates when used in difficult clinical settings, such as in patients with severe sepsis. In this way, antibiotic de-escalation based on the 2017 Japanese Respiratory Society guidelines led to a reduction in total antibiotic costs for the management of community-acquired non-bacteremic pneumococcal pneumonia [38].

In the same way, the treatment duration was excessive or doubtful in 55% of reviewed cases.

The duration of antibiotic therapy for most bacterial infections is based on the fact that the week has 7 days in it, resulting in traditional 7- to 14-day antibiotic courses. So, the weekly period determines the therapy duration [39]. The proposal to reduce the duration of the treatment of antibiotics is based on the rapid effect of antibiotics (clinically visible in the first 3–4 days and with maximum efficacy 5–8 days after onset). At the same time, the selection of resistant mutants starts from 3–4 days after the start of treatment and increases as time increases [40]. In fact, excess treatment has not been associated with lower rates of any adverse outcomes, including death, readmission, emergency department visit, or Clostridioides difficile infection. Each excess day of treatment was associated with a 5% increase in the odds of antibiotic-associated adverse events reported by patients after discharge. The percentage of patients (±SD) who received excess treatment varied from 38.1% ± 3.7% to 95.0% ± 2.3% among hospitals [32]. In our hospital, it was 45.45% of the patients. In a systematic review and meta-analysis [41] about the efficacy of short duration in CAP, short-course antibiotic treatment (≤6 days) was as effective as and potentially superior to, in terms of mortality and serious adverse events, longer-course treatment. In the several RCTs that evaluated the effect on resistance [42], shorter courses decreased the emergence of antibiotic resistance in respiratory secretions. For instance, children receiving 5 days of beta-lactam therapy for CAP had a significantly lower abundance of antibiotic resistance determinants than those receiving standard 10-day treatment [43].

In conclusion, it is necessary to optimize the management of antibiotic treatment in the analyzed departments. Nowadays there is a wide range of treatment guides responsible for clinicians instead of one standardized line of treatment which provides several different options of therapy choices and durations. The institution of one unique guide based on knowledge and clinical research will provide the right path to achieve and standard prescription of antibiotics therapy and its length and de-escalation if it is necessary.

The objective of changing actual antibiotic use requires both the knowledge of the current treatment situation and the appropriate treatment criteria guidelines. The proposed action plan is divided into several phases. The awareness of current antibiotic use will be disseminated in professional monographic sessions. The correct pneumonia treatment and antibiotics use will be explained in theory classes taught by experts in the field to widespread the selected guide and clinical keys focused on the early mobilization of patients. [22]. At the same time, audits and feedback will be proposed through an antibiotic stewardship program: a prospective audit with intervention and feedback involves the assessment of antimicrobial therapy by trained individuals (usually physicians and/or pharmacists), who make recommendations to the prescribing service in real-time when therapy is considered suboptimal. Besides this, defining stability criteria to apply sequential therapy/de-escalation in those monographic sessions was considered essential.

## Figures and Tables

**Figure 1 antibiotics-11-01336-f001:**
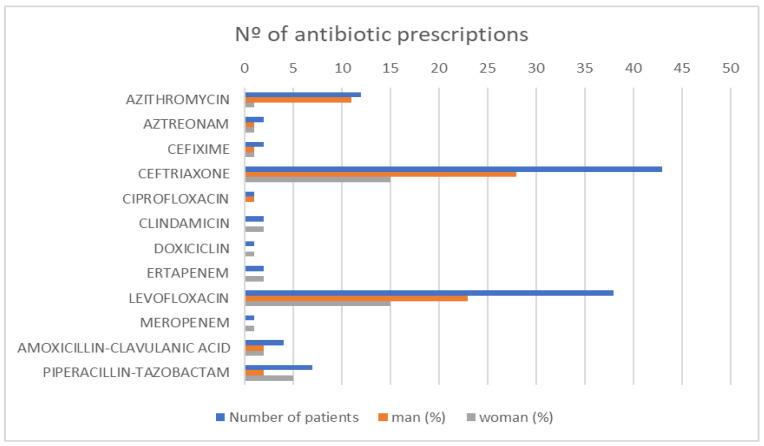
The number of antibiotics prescribed.

**Table 1 antibiotics-11-01336-t001:** An abbreviated overview of the clinical practice-guideline recommendations of reference hospitals.

Comunity-Acquired Pneumonia (CAP)	Drug	Duration
**Non-severe (PSI I–II)**	Amoxicilina 1 g/8 h PO Amoxicilina-clavulanic acid 875/125 mg/8 h PO Azythromicin * 500 mg PO (atypical)	5–7 days
**Severe (PSI III)**	ceftriaxone 2 g (pneumococcal antigenuria positive)	5–7 days
	ceftriaxone 2 g + (azythromicine * 500 mg or levofloxacin 500 mg)	7–10 days
**Severe (PSI IV–V)**	ceftriaxone 2 g (or 1 g/12 h) + (azythromicine * 500 mg or levofloxacin 500 mg)	7–10 days

* Azythromycin 3 days, PSI: Pneumonia Severity Index.

**Table 2 antibiotics-11-01336-t002:** Charlson Index, PSI scale and mean age.

Variable	Values	Total Number (%)	Male (%)	Female (%)
**Charlson index**	0	20 (36.36)	13 (39.39)	7 (31.82)
	[1,2,3]	20 (36.36)	11 (33.33)	9 (40.91)
	>3	14 (25.45)	8 (24.24)	6 (27.27)
	Not registered	1 (1.82)	1 (3.03)	0 (0.00)
**PSI scale**	I	9 (16.36)	5 (15.15)	4 (18.18)
	II	10 (18.18)	9 (27.27)	1 (4.55)
	III	9 (16.36)	6 (18.18)	3 (13.64)
	IV	16 (29.09)	8 (24.24)	8 (36.36)
	V	2 (3.64)	2 (6.06)	0 (0.00)
	Not registered	9 (16.36)	3 (9.09)	6 (27.27)
**Mean age (years ± SD)**		65.07 ± 22.12	66.06 ± 21.20	69.58 ± 23.18

PSI: Pneumonia Severity Index; SD: standard deviation.

**Table 3 antibiotics-11-01336-t003:** Results for treatment process consensus about considered items.

Variable	Values	Total (%)	Male (%)	Female (%)
**Indication of the antibiotic**	**Adequate**	53 (96.36)	31 (93.94)	22 (100.00)
	**Doubtful**	1 (1.82)	1 (3.03)	0 (0.00)
	**Inadequate**	1 (1.82)	1 (3.03)	0 (0.00)
**Choice of antibiotic**	**Optimal**	35 (63.64)	20 (60.61)	15 (68.18)
	**might be better**	14 (25.45)	9 (27.27)	5 (22.73)
	**Inadequate**	1 (1.82)	1 (3.03)	0 (0.00)
	**Doubtful**	5 (9.09)	3 (9.09)	2 (9.09)
**Time of administration of the first dose**	**Adequate**	50 (90.91)	29 (87.88)	21 (95.45)
	**Doubtful**	5 (9.09)	4 (12.12)	1 (4.55)
**De-escalation/ oral sequencing**	**Optimal**	23 (41.82)	13 (39.39)	10 (45.45)
	**might be better**	28 (50.91)	19 (57.58)	9 (40.91)
	**Doubtful**	4 (7.27)	1 (3.03)	3 (13.64)
**Treatment duration**	**Optimal**	25 (45.45)	15 (45.45)	10 (45.45)
	**Excessive**	25 (45.45)	14 (42.42)	11 (50.00)
	**Doubtful**	5 (9.09)	4 (12.12)	1 (4.55)
**Monitoring of efficacy and adverse effects**	**Adequate**	53 (96.36)	31 (93.94)	22 (100.00)
	**Doubtful**	2 (3.64)	2 (6.06)	0 (0.00)
**Registration in the medical record**	**Complete**	19 (34.55)	10 (30.30)	9 (40.91)
	**Partial**	32 (58.18)	20 (60.61)	12 (54.55)
	**Lacking**	4 (7.27)	3 (9.09)	1 (4.55)
**Total**		55 (100.0)	33 (60.0)	22 (40.0)

The data in blue were the most striking data for the team.

**Table 4 antibiotics-11-01336-t004:** Distribution of treatment duration.

PSI	Number of Patients	Mean	Standard Deviation	Confidence Interval
**PSI I–II**	**19**	13.21	6.45	(10.10–16.32)
**PSI III**	**9**	10.78	3.38	(8.18–13.38)
**PSI IV**	**16**	12.56	3.41	(10.75–14.38)
**PSI V**	**2**	8.50	2.12	(−10.56–27.56)
**Missing PSI values**	**9**	NA	NA	NA

PSI: Pneumonia Severity Index; NA: not applied.

## Data Availability

Not applicable.

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
