# Peer review of "Multidisciplinary Collaboration for the Optimization of Antibiotic Prescription: Analysis of Clinical Cases of Pneumonia between Emergency, Internal Medicine, and Pharmacy Services"

_antibiotics, 2022, doi:10.3390/antibiotics11101336_

Round 1
Reviewer 1 Report
Dear Authors,
The article addresses a topic of major importance at the current moment when there is an alarming escalation of antibiotic resistance of microorganisms, the need for optimal therapeutic strategies being more and more evident.
In order to have a clearer picture of the topic addressed, I suggest some corrections or approaches to improve the presented study:
Line 79: The authors emphasize the existence of numerous studies to highlight the multidisciplinary collaboration that can increase the strategies for prescribing, administering, etc. of antibiotics. More examples are needed considering that only one bibliographic title was cited ([20]).
Lines 97-99: The study was conducted in a hospital unit with 72 beds. The authors should provide additional information on the inclusion/exclusion criteria regarding the period under study, which is quite short (January-May 2021) and the geographic region: it is a correlation with the COVID-19 pandemic situation, related to the period and the region?
Line 138: The evaluation criteria of the applied treatment are highlighted. The item "choice of antibiotic" in this study highlights that the approach from different guidelines was followed exclusively, the personalization of the treatment, the specific approach of testing the sensitivity of the microorganism is not mentioned as a criterion. I suggest the authors to include this extremely important aspect.
Lines 211-212: The study refers to the period January-May 2021. For better clarity, the authors should reformulate the phrase "57 patients remain alive in May 2022" from which it can be understood on the one hand that the rest of patients are no longer alive, and on the other hand, that the study was completed in May 2022 (?).
In the discussion section there are some aspects that can be improved:
Lines 260-262: Given that the strategy in the therapeutic approach with antibiotics is extremely important, the bibliographic references should be expanded, [25] from 2013 should also be improved with other more recent examples from the specialized literature.
Line 263: The authors should develop the proposal of the three strategies by adding some therapeutic alternatives based on the data obtained from the multidisciplinary collaboration (microbiologists, clinical pharmacist, etc.) and personalized therapies
Lines 286-288: The trend of using antibiotics is highlighted, but the question can be asked where the personalized antibiotic treatment is placed to limit the emergence of resistance
Author Response
Line 79: The authors emphasize the existence of numerous studies to highlight the multidisciplinary collaboration that can increase the strategies for prescribing, administering, etc. of antibiotics. More examples are needed considering that only one bibliographic title was cited ([20]). We have put 13,14 but in the bibliography several appear: 18,26
Lines 97-99: The study was conducted in a hospital unit with 72 beds. The authors should provide additional information on the inclusion/exclusion criteria regarding the period under study, which is quite short (January-May 2021) and the geographic region: it is a correlation with the COVID-19 pandemic situation, related to the period and the region? Yes, these are the moths with the major incidence of pneumonia in basque country in the north of Spain.
Line 138: The evaluation criteria of the applied treatment are highlighted. The item "choice of antibiotic" in this study highlights that the approach from different guidelines was followed exclusively, the personalization of the treatment, the specific approach of testing the sensitivity of the microorganism is not mentioned as a criterion. I suggest the authors to include this extremely important aspect. We were unable to get results of sputum culture but yes streptococcus and legionella antigen test.
Lines 211-212: The study refers to the period January-May 2021. For better clarity, the authors should reformulate the phrase "57 patients remain alive in May 2022" from which it can be understood on the one hand that the rest of patients are no longer alive, and on the other hand, that the study was completed in May 2022 (?). The study was retrospective in those months and it was carried out in may 2022 (not finished), It was finished in may 2021.
In the discussion section there are some aspects that can be improved:
Lines 260-262: Given that the strategy in the therapeutic approach with antibiotics is extremely important, the bibliographic references should be expanded, [25] from 2013 should also be improved with other more recent examples from the specialized literature. 25,26,27 and 28 is all around the strategies. We have changed the 28th reference from 2012 reference to the actual reference published in spring of 2022 (PROA-2) in Spain. The use of procalcitonin may be useful but more evidence is needed.
Line 263: The authors should develop the proposal of the three strategies by adding some therapeutic alternatives based on the data obtained from the multidisciplinary collaboration (microbiologists, clinical pharmacist, etc.) and personalized therapies . We have done once we see our shortcomings: we have a guideline but most of people used other guides and it was not so clear stability criteria and the excessive treatment was for fear of not being enough, they were unaware of the new evidence. We have to do educational interventions
Lines 286-288: The trend of using antibiotics is highlighted, but the question can be asked where the personalized antibiotic treatment is placed to limit the emergence of resistance. Educational intervention is needed
Reviewer 2 Report
This retrospective observational study evaluated the appropriateness of antimicrobial use for 55 community acquired pneumonia (CAP) patients over a 5-month period in a small hospital in Spain. The study delivered institutional data on indication, choice of antibiotic, timing of first dose, antibiotic de-escalation and intravenous (IV)-to-oral (PO) antibiotic switch, and duration, etc.
--Line 55-57, the reference quoted data from 2009-2013, is there any more recent data?
--Line 118, “treatment revision procedure”, it would be helpful to elaborate briefly on what’s treatment revision procedure.
--Line 223-224, it’s author’s first time mentioned “FINE I-II” in the text, it would be helpful to define “FINE I-II” in method section.
--Under result section, I would appreciate authors to include data on defined daily dose (DDD) or DDD/100 admission or DDD/100 beds, and clinical outcome such as mortality data.
--Tables 2, 3, 4, columns can be centered.
--Table 3, treatment duration. In addition to overall duration, it would be helpful to separate duration of non-severe and severe CAP since the recommended duration is longer for severe CAP according to the institutional guideline. And please include discussion on possible reasons of longer duration was observed in 45.5% patients.
--Line 299, there is a typo “16,77%”, change to “16.77%).
--Under discussion, it would be great to include a plan of action or specific stewardship interventions on how you would like to improve antibiotic prescribing in addition to share data with various hospital disciplines since there are lots of room for improvement based on the finding of antibiotic choice, de-escalation/IV-to-PO switch, especially duration of therapy.
--This retrospective review served as a benchmark or baseline data on the appropriateness of antibiotic use for CAP at IHO. It would be more interesting for authors to do a performance improvement project to compare pre- and post- stewardship or research team intervention on antibiotic choice, de-escalation/IV-to-PO switch, and DDD since this data is collected in Jan-May 2001.
Author Response
--Line 55-57, the reference quoted data from 2009-2013, is there any more recent data? Yes [7].
--Line 118, “treatment revision procedure”, it would be helpful to elaborate briefly on what’s treatment revision procedure. I don´t understand because it´s detailed. I have put the criteria in black to see it better and we have tried to explain better
--Line 223-224, it’s author’s first time mentioned “FINE I-II” in the text, it would be helpful to define “FINE I-II” in method section. FINE=PSI, We have changed, sorry.
--Under result section, I would appreciate authors to include data on defined daily dose (DDD) or DDD/100 admission or DDD/100 beds, and clinical outcome such as mortality data. In suplementary data is DDD/100 beds and DDD/100 admission in internal medicine and DDD/1000 visits in emergency. We have explained better just in case. Mortality data 35/90*100= 38,89%. We have added it.
--Tables 2, 3, 4, columns can be centered. Changed
--Table 3, treatment duration. In addition to overall duration, it would be helpful to separate duration of non-severe and severe CAP since the recommended duration is longer for severe CAP according to the institutional guideline. And please include discussion on possible reasons of longer duration was observed in 45.5% patients. Done
--Line 299, there is a typo “16,77%”, change to “16.77%”. We have changed, sorry.
--Under discussion, it would be great to include a plan of action or specific stewardship interventions on how you would like to improve antibiotic prescribing in addition to share data with various hospital disciplines since there are lots of room for improvement based on the finding of antibiotic choice, de-escalation/IV-to-PO switch, especially duration of therapy. We have made better, we hope you like it
--This retrospective review served as a benchmark or baseline data on the appropriateness of antibiotic use for CAP at IHO. It would be more interesting for authors to do a performance improvement project to compare pre- and post- stewardship or research team intervention on antibiotic choice, de-escalation/IV-to-PO switch, and DDD since this data is collected in Jan-May 2021. That project we will do the next year
Round 2
Reviewer 2 Report
Thank you for providing the updated references and more information on DDD in supplement documents.